# Origin of the quasi-quantized Hall effect in ZrTe$_5$

S. Galeski [1✉], T. Ehmcke [2], R. Wawrzyńczak[1], P. M. Lozano[3], K. Cho[4], A. Sharma[4], S. Das[4], F. Küster[4], P. Sessi [4], M. Brando[1], R. Küchler[1], A. Markou [1], M. König[1], P. Swekis[1], C. Felser[1], Y. Sassa[5], Q. Li [3], G. Gu[3], M. V. Zimmermann [6], O. Ivashko [6], D. I. Gorbunov[7], S. Zherlitsyn[7], T. Förster[7], S. S. P. Parkin [4], J. Wosnitza[7,8], T. Meng [2] & J. Gooth [1,8✉]

The quantum Hall effect (QHE) is traditionally considered to be a purely two-dimensional (2D) phenomenon. Recently, however, a three-dimensional (3D) version of the QHE was reported in the Dirac semimetal ZrTe$_5$. It was proposed to arise from a magnetic-field-driven Fermi surface instability, transforming the original 3D electron system into a stack of 2D sheets. Here, we report thermodynamic, spectroscopic, thermoelectric and charge transport measurements on such ZrTe$_5$ samples. The measured properties: magnetization, ultrasound propagation, scanning tunneling spectroscopy, and Raman spectroscopy, show no signatures of a Fermi surface instability, consistent with in-field single crystal X-ray diffraction. Instead, a direct comparison of the experimental data with linear response calculations based on an effective 3D Dirac Hamiltonian suggests that the quasi-quantization of the observed Hall response emerges from the interplay of the intrinsic properties of the ZrTe$_5$ electronic structure and its Dirac-type semi-metallic character.

[1] Max Planck Institute for Chemical Physics of Solids, Dresden, Germany. [2] Institute for Theoretical Physics and Würzburg-Dresden Cluster of Excellence ct. qmat, Technische Universität Dresden, Dresden, Germany. [3] Condensed Matter Physics and Materials Science Department, Brookhaven National Laboratory, Upton, NY, USA. [4] Max Planck Institute of Microstructure Physics, Halle, Saale, Germany. [5] Department of Physics, Chalmers University of Technology, Gothenburg, Sweden. [6] Deutsches Elektronen-Synchrotron DESY, Hamburg, Germany. [7] Hochfeld-Magnetlabor Dresden (HLD-EMFL) and Würzburg-Dresden Cluster of Excellence ct.qmat,, Helmholtz-Zentrum Dresden-Rossendorf, Dresden, Germany. [8] Institut für Festkörper- und Materialphysik, Technische Universität Dresden, Dresden, Germany. ✉email: stanislaw.galeski@cpfs.mpg.de; johannes.gooth@cpfs.mpg.de

Electrons subject to a magnetic field $B$ are forced to move on curved orbits with a discrete set of energy eigenvalues—the Landau levels (LLs). By increasing $B$, the LLs shift through the Fermi level $E_F$ one after the other, leading to quantum oscillations in transport and thermodynamic quantities[1]. At sufficiently large magnetic field, where only a few LLs are occupied, 2D electron systems (2DESs) enter the quantum Hall regime[2–5]. This regime is characterized by a fully gapped electronic spectrum in the bulk and current-carrying gapless edge states, leading to quantization of the Hall conductance $G_{xy} = v e^2/h$, where $v$ is the Landau-level filling factor, $e$ is the elementary charge, and $h$ is Planck's constant. The situation is different in three dimensions. Instead of fully gapping the bulk of the 3D electron gas, high magnetic fields confine the electron motion in the plane perpendicular to the magnetic field still allowing them to freely move along the field, making the electron motion one-dimensional. Hence, current flow is allowed in the direction parallel to $B$. Crucially, the missing gap spoils quantization of $G_{xy}$. However, it is predicted that a 3D version of the QHE could occur in semimetals and doped semiconductors[6–9], in which the application of a magnetic field would lead to a Fermi surface instability, causing a periodic modulation of the electron density along the direction of $B$. Such modulation can effectively be thought of as a stack of 2DESs, each layer being in the quantum Hall regime. The signature of such a 3D quantum Hall system is that the Hall conductivity exhibits plateaus of $\sigma_{xy} = v e^2/h \cdot G_z/2\pi$ that are accompanied by a vanishing longitudinal electrical conductivity $\sigma_{xx}$[6]. $G_z$ is the reciprocal lattice vector of the modulation along $B$.

Recently, a quantized Hall response has been observed in the prototypical 3D Dirac semimetal[10–13] materials $ZrTe_5$[14] and $HfTe_5$[15,16]. Particularly, the Hall resistivity $\rho_{xy}$ has been found to exhibit a plateau, scaling with $(e^2/h \cdot k_{F,b}/\pi)^{-1}$ for the magnetic field aligned with the crystal's $b$-axis when only the last LL is occupied (quantum limit). $k_{F,b}$ is the Fermi wave vector along the crystal $b$-axis at zero magnetic field. The observed scaling of the plateau height with $k_{F,b}/\pi$ has been interpreted as indicating a correlation-driven origin of the Hall effect. It was suggested[14] that the applied magnetic field leads to a Fermi surface instability and formation of a charge-density wave (CDW) with a wavelength of $\lambda_z = \pi/k_{F,b}$ along the magnetic field. In a CDW, the density of the electrons and the position of the lattice atoms are periodically modulated with a wavelength, usually much larger than the original lattice constant[17]. CDWs are usually the energetically preferred ground state of interacting quasi-one-dimensional conductors due to the almost perfect nesting of the Fermi surface[17]. In 3D systems, the dimensional reduction of the energy spectrum in high magnetic fields, in principle supports a scenario of a field-induced CDW, in which each CDW modulation "layer" contributes the conductance of one 2D quantum Hall system to the total bulk Hall conductivity.

The main argument for the CDW origin of the Hall plateaus in ref. [15] is the vanishing longitudinal resistance in the quantum limit, which was interpreted as a consequence of a fully gapped electronic structure in the bulk and non-dissipative edge channels in each electron layer. However, the longitudinal resistivity seems to remain finite in the majority of the $ZrTe_5$[14] and $HfTe_5$[15,16] samples despite the presence of pronounced plateaus in the Hall resistivity (even down to 50 mK). Thus, it is desirable to investigate the physics of $ZrTe_5$ in magnetic fields beyond simple charge transport experiments.

In transport, a CDW transition typically manifests as an abrupt increase of electrical resistance due to the gapping of the Fermi surface and non-ohmic transport characteristics[17]. However, a CDW ground state cannot be identified by transport measurements alone, as such features also exist in materials without a CDW transition[18–22]. Both gapping of the Fermi surface and

emergence of a periodic charge modulation should clearly impact thermodynamic, structural, and spectral properties[1]. Gap openings in the electronic density of states (DOS) in non-magnetic materials can, for example, be probed by magnetization measurements[23], thermoelectric coefficients[24], and scanning tunneling spectroscopy[17]. The corresponding periodic charge modulation typically leads to a change in the phonon spectrum, which can be directly seen in Raman scattering[17], X-ray diffraction[25,26], and ultrasound propagation measurements[27,28]. In addition, Raman spectroscopy can directly probe the Raman active CDW amplitude mode[17].

## Results

**Experimental results**. In our study, we have used similar $ZrTe_5$ samples as those studied in ref. [14] grown by tellurium flux method and applied all the after-mentioned experimental techniques (see "Methods" for details). In total, we have investigated over 20 samples from which we exemplarily show data from Samples A, B, C, D, E, F, G, and H, all exhibiting consistent results (Supplementary Table 1). Upon cooling in $B = 0$ T, the longitudinal resistivity $\rho_{xx}$ increases with decreasing temperature $T$ until reaching a maximum at $T_L = 90$ K (Fig. 1a, Supplementary Fig. 1 and ref. [14]). This maximum has been previously observed in $ZrTe_5$ and has been attributed to a Lifshitz transition that is accompanied by a change of charge-carrier type[29]. Consistently, the sign of the zero-field Seebeck coefficient $S_{xx}$ changes sign at $T_L$[11,30], indicating electron-type transport for $T < T_L$ and hole-type transport above.

Similar to previous studies[14,16], we find Shubnikov-de Haas oscillations in $\rho_{xx}(B)$ at low temperatures ($T < 30$ K) that are consistent with a single ellipsoidal Fermi (Fig. 1b). The magnetotransport measurement configuration is sketched in Fig. 1c. Further analysis of the Shubnikov-de Haas oscillations reveal that the quantum limit is reached at magnetic fields as low as $B_{QL} = 1.8$ T for the magnetic field applied along the $b$-axis (Supplementary Figs. 3, 6, 7, and 14) and a relatively small effective mass of the order of 0.01 $m_0$ (Supplementary Table 1), which is in agreement with Dirac fermions ($m_0$ is the bare electron mass)[13]. Importantly, in this field configuration (see for example Fig. 1b), all studied samples show a pronounced plateau in $\rho_{xy}(B)$ in the quantum limit with a height close to $(h/e^2)\,\pi/k_{F,b}$ as reported in ref. [14]. However, $\rho_{xx}(B)$ does not vanish in any of the measured samples, even when cooled to 50 mK[16], consistent with the Landau quantization of a 3D Fermi pocket. While $\rho_{xx}(B)$ remains finite in all samples, it is always much smaller than $\rho_{xy}(B)$ at low temperatures and thus the Hall conductivity $\sigma_{xy} = \rho_{xy}/(\rho_{xx}^2 + \rho_{xy}^2)$ reduces to $\sigma_{xy} \approx 1/\rho_{xy}$, enabling the observation of the quantization in $\rho_{xy}$.

To test the hypothesis of a field-induced CDW, we have first investigated $S_{xx}(B)$ at low temperatures. $S_{xx}$ is particularly sensitive to changes in the electronic DOS, i.e., gapping of the Fermi surface, because it is proportional to the energy derivative of the DOS at $E_F$ (see Supplementary Information and ref. [31] for a detailed discussion). However, we do not find signatures of a CDW in the $B$-dependent $S_{xx}$ with $B$ applied along with the $a$, $b$, and $c$ crystallographic axes and for various angles in between them. Instead, the data show quantum oscillations (Fig. 1d–f and Supplementary Figs. 4 and 9) that are consistent with a single ellipse-shaped 3D electron pocket at $E_F$ (Fig. 1g–i) as observed in $\rho_{xx}(B)$.

Next, we have measured the magnetization of $ZrTe_5$ across the hypothetical phase boundary proposed in ref. [14]. Magnetization is a thermodynamic quantity with the paramagnetic contribution directly proportional to the DOS and hence, highly sensitive to phase transitions[23], such as FS gapping due to the formation of a

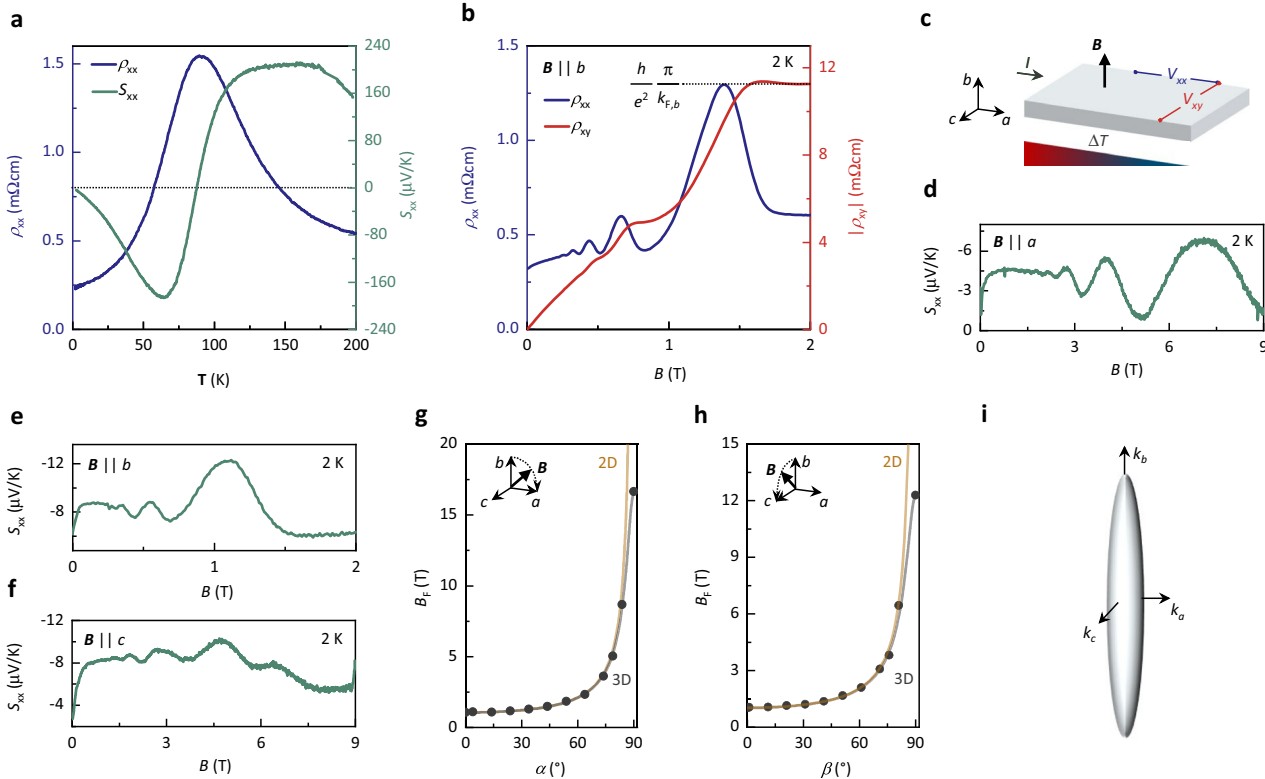

**Fig. 1 Three-dimensional morphology of the Fermi surface and quasi-quantized Hall effect in ZrTe$_5$. a** Longitudinal electrical resistivity $\rho_{xx}$ and Seebeck coefficient $S_{xx}$ of Sample A as a function of temperature $T$ at zero magnetic field. **b** $\rho_{xx}$ and Hall resistivity $\rho_{xy}$ as a function of $B$ applied along the $b$-axis at 2 K, obtained on Sample B. The last Hall plateau scales with the experimentally extracted Fermi wave vector $k_{F,b}$ along $B$ (Supplementary Table 1), the electron charge $e$ and the Planck constant $h$. **c** Sketch of the transport measurement configurations with respect to the three crystal axes $a$, $b$, and $c$. The electrical current $I$ and the temperature gradient $\Delta T$ are applied along the $a$-axis. The corresponding longitudinal ($V_{xx}$) and Hall ($V_{xy}$) voltage responses are measured along the $a$-axis ($V_{xx}$) and along $c$-axis, respectively. **d** $S_{xx}$ as a function of $B$ at 2 K with $B$ applied along the $a$, **e**, along the $b$ and **f**, along the $c$-axis, measured on Sample A. **g** Shubnikov-de Haas frequency $B_F$ as a function of angle ($\alpha$ and $\beta$) between $B$ and the $b$-axis, rotated within the $a–b$ plane and (**h**), within the $b–c$ plane of Sample A. The black dots represent the measurement data. The yellow lines represent the fitting curves of a planar 2D Fermi surface model to the data. The black lines represent fitting curves of an ellipsoidal 3D Fermi surface model to the data. **i** Sketch of the experimentally extracted Fermi surface of ZrTe$_5$ along the momentum vectors $k_a$, $k_b$ and $k_c$ in $a$, $b$, and $c$ direction, respectively.

CDW (see Supplementary Note 6 for a detailed discussion). Figure 2a and b shows the temperature dependence of magnetization measured at 2 T and 100 mT with the field applied along the $b$-axis—the field configuration in which the Hall plateaus are seen. The two investigated field values are chosen to cover two regimes: at 2 T the system is in the quantum limit and at 100 mT multiple LLs are occupied. In both regimes, we find that the magnetization does not show any signatures of the formation of a CDW. Investigation of the magnetization as a function of magnetic field (Fig. 2d and Supplementary Fig. 5) at low temperature reveals pronounced de Haas-van Alphen oscillations on top of a roughly linear background. The de Haas-van Alphen oscillations are in agreement with the Landau quantization of a single Fermi pocket and, hence, consistent with the observations in $\rho_{xx}(B)$ and $S_{xx}(B)$[32]. The strong linear diamagnetic background is indicative of a small effective mass, as expected for Dirac Fermions[32,33].

Independently from transport and thermodynamic probes, the gap opening in the electronic structure, related to the formation of a CDW, could be probed using scanning tunneling spectroscopy (STS), since STS provides direct information about the DOS as a function of energy. Therefore, STS would directly corroborate the presence of a field-induced gap tied to the formation of a CDW[17]. We have performed STS experiments on our ZrTe$_5$ samples at 0.4 K in fields up to 5 T applied along the crystallographic $b$-axis (Fig. 2e and Supplementary Fig. 6). The

measurements did not reveal any features in the density of states that could be attributed to a CDW formation. The DOS at E$_F$ remained finite even deep in the quantum limit, consistent with the observed Landau quantization of a 3D Fermi surface in $\rho_{xx}(B)$, $S_{xx}(B)$, and $M(B)$ and finite longitudinal resistivity. The observed band edges (labeled as CB$_1$, CB$_2$, and VB$_1$ in Fig. 2e) are consistent with previous experiments on ZrTe$_5$[34] and remain fixed with respect to each other—a characteristic feature of the lowest LLs of Dirac fermions[35].

In addition, we have performed ultrasound propagation and attenuation experiments. Measurements of the sound-velocity probe the system's elastic modulus, and thus a thermodynamic quantity that is very sensitive to sudden changes in the free energy, such as those caused by a CDW gap opening[27,28,36]. Ultrasound propagation measurements can provide information about the electron–phonon coupling, a quantity crucial for the formation of a CDW (see "Methods" and Supplementary Note 8 for a detailed discussion). Figure 2f and g shows the variation of the sound velocity $\Delta v_s/v_s$ and the sound attenuation $\Delta \alpha$ of the longitudinal sound modes (propagation along the $a$-axis and longitudinal polarization vector along the $a$-axis) as a function of the magnetic field applied along the $b$-axis at 2 K. We find that both $\Delta v_s/v_s$ and $\Delta \alpha$ do not exhibit any anomalies that could indicate a phase transition. Instead, $\Delta v_s/v_s$ and $\Delta \alpha$ consistently reflect the quantum oscillations observed in $\rho_{xx}(B)$, $S_{xx}(B)$, and $M(B)$ that stand for a single Fermi pocket at E$_F$ (see also

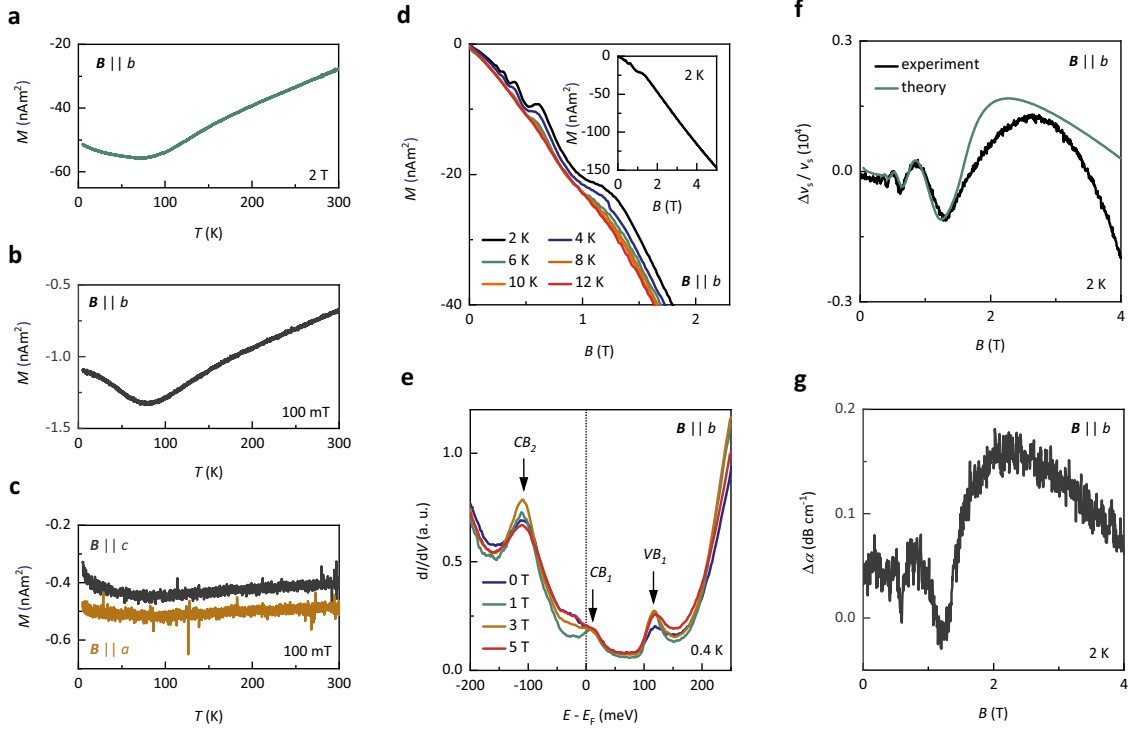

**Fig. 2 Probing the electronic spectrum of ZrTe₅ in magnetic fields. a** Magnetization $M$ as a function of temperature $T$ with a 2 T magnetic field $B$ applied along the $b$-axis of Sample C. **b** $M$ as a function of $T$ with a 50 mT applied along the $b$-axis of Sample C. **c** $M$ as a function of $T$ with a 100 mT applied along the $a$-axis (black line) and $c$-axis (dark yellow line) of Sample C. **d** De Haas-van Alphen oscillations observed in $M$ as a function of $B$ applied along the $b$-axis at various $T$ on Sample C. The inset shows a zoom-out of $M$ as a function of $B$ applied along the $b$-axis at 2 K. **e** Differential conductance $dI/dV$ as a function of bias voltage (energy $E$) with respect to the Fermi level $E_F$ (dotted line) for various $B$ applied along the $b$-axis of Sample D at 0.4 K. The conduction band edges are marked with $CB_1$ and $CB_2$, the valence band edge is marked with $VB_1$. **f** Sound-velocity variation $\Delta v_s/v_s$ of the longitudinal sound mode (propagation along the $a$-axis) as a function of the magnetic field applied along the $b$-axis at 2 K on Sample E. The black line is experimental data, the green line a theoretical model (see "Methods"), based on a Dirac Hamiltonian. **g** Sound attenuation $\Delta\alpha$ of the longitudinal mode (propagation along the $a$-axis, polarization vector along the $a$-axis) as a function of the magnetic field applied along the $b$-axis at 2 K on Sample E. None of the experiments shows signatures of a charge-density–wave formation.

Supplementary Fig. 5 for the transverse mode) It has been pointed out[37] that for a CDW to emerge in the quantum limit regime above 1.5 T, the strength of the effective electron–electron interaction generated from electron–phonon coupling should be around $g_0 = 537.3$ eV/nm. Our analysis of the quantum oscillations in $\Delta v_s/v_s$, however, suggests a much smaller coupling constant $g_0 = 0.015$ eV/nm (see "Methods"). Since $g_0$ depends quadratically on the electron–phonon coupling, we find that the electron–phonon-coupling in ZrTe₅ is too small by two orders of magnitude for a CDW to appear.

A field-induced phase transition into a CDW ground state is expected to lead to a modification of the crystal structure due to the new emerging charge modulation. To check if this is the case in ZrTe₅ we have performed X-ray diffraction measurements at zero field and in the quantum limit at 2 K. The formation of a CDW is expected to lead to the emergence of satellite super-structure peaks in the X-ray spectra with corresponding $k$-vectors of $G_z = 2k_{F,b}$ adjacent to the main Bragg reflections[17]. The results of Q-scans along the $b$-direction of the crystal in the vicinity of the (010)-Bragg peak with the field aligned along the $b$-axis are shown in Fig. 3a. We find that the (010) Bragg peak does not change up to 2 T, and—most importantly—does not show any emerging satellite reflections[26]. The X-Ray experiments indicate that the zero-field crystal structure of ZrTe₅[38] is maintained in finite fields up to the quantum limit.

This result is confirmed by the magnetic field-dependent Raman spectroscopy measured at 2 K. The formation of CDW

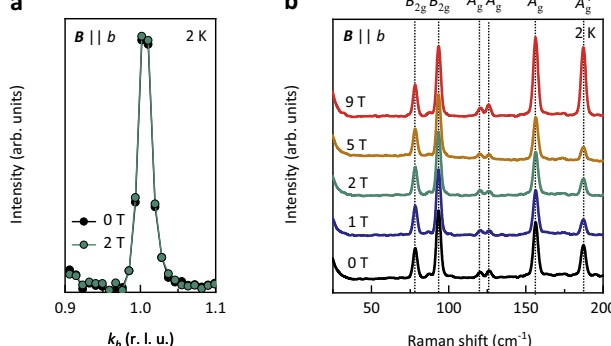

**Fig. 3 Probing the crystal lattice of ZrTe₅ in magnetic fields. a** The (010)-peak observed in X-Ray diffraction on Sample A at magnetic fields $B$ of 0 T and 2 T at 2 K. **b** Raman spectra of Sample F for various $B$ at 2 K. For fields up to 9 T, the Raman modes are located at 67, 84, 113, 118, 144, and 178 cm⁻¹, labeled with $B^1_{2g}$, $B^2_{2g}$, $A^1_g$, $A^2_g$, $A^3_g$, and $A^4_g$, respectively. None of the experiments shows signatures of a charge-density–wave.

would cause additional peaks[17,39] to appear in the Raman spectra of ZrTe₅[40,41]. However, apart from small changes in the amplitude of the vibration modes, the Raman spectra do not change upon application of magnetic fields up to 9 T (Fig. 3b). In particular, the existing phonon modes do not show the change in

Raman shift and no new spectral features appear that could be related to absorption by the Raman active CDW amplitude mode.

Our experimental investigations indicate that the state underlying the emergence of Hall plateaus in the quantum limit of ZrTe$_5$ is in fact gapless and exhibits the behavior of a Fermi sea of a Landau-quantized 3D Dirac semimetal.

**Theoretical results**. To directly compare the effect of Landau quantization of a 3D Dirac system on the Hall resistivity, we performed linear response calculations of the electrical transport properties, based on an anisotropic Dirac Hamiltonian, with a magnetic field along the $z$-direction (see "Methods"). As a cross-check, we have also computed $S_{xx}$, $S_{xy}$, $\Delta v_s/v_s$, and $M$ (see "Methods" for details), using the same model. Although the inverse Landau-level broadening and the transport relaxation time are in general sensitive to different scattering mechanisms, the calculations were performed assuming that both quantities are the same. This approximation has the advantages of being transparent to interpretation and that the calculated quantities can be directly related to the band structure. However, as a side effect, the calculated $\rho_{xx}$ is slightly different than seen in the experiment (Fig. 4). More importantly, our model reproduces all other experimental data even on a quantitative level (Figs. 2 and 4) and explains the key observations of the electrical transport in ZrTe$_5$: first, at high magnetic fields, Landau quantization leads to plateaus in $\rho_{xy}(B)$ with a height of $(h/e^2)\,\pi/k_{F,b}$ in the quantum limit. For a 3D system, the Landau levels form continuous Landau bands dispersing parallel to the applied field. The Hall conductivity is the sum of conductance quanta over all occupied Landau bands $\nu$ and wave numbers $k_b$,

$$\sigma_{xy}(B) = \sum_{\nu \, occ.} \int_{k_b \, occ.} \frac{dk_b}{2\pi} \frac{e^2}{h} = \sum_{\nu \, occ.} \frac{2k_{F,b,\nu}(B)}{2\pi} \frac{e^2}{h},$$

where $k_{F,\nu,b}(B)$ is the

Fermi wave number parallel to the field. While $k_{F,b,\nu}$ in general depends on $B$, a characteristic feature of Dirac systems is that the lowest LL does not shift with magnetic field[35] (see Supplementary Fig. 16) and $\sigma_{xy}(B)$ becomes $\frac{2k_{F,b}}{2\pi}\frac{e^2}{h}$ in the quantum limit. Second, while $\rho_{xx}(B)$ shows Shubnikov-de Haas oscillations with minima at the center of the Hall plateaus, it is finite for all magnetic fields, due to the gapless Landau band structure.

Magnetotransport in 2DESs with localized states is often described by effectively fixed chemical potentials, which leads to well-defined quantum Hall plateaus. In contrast, the large carrier densities $n$ in ordinary 3D metals, such as copper, usually forces the particle number to be conserved in order to avoid large charging energies. This implies variation of the chemical potential as a function of magnetic field. In materials with small Fermi surfaces, such as Dirac semimetals, the charging energies on the contrary remain relatively small, since the absolute change in $n$ with $B$ at fixed $E_F$ remains small. Defects can furthermore induce localized states that absorb some of the conduction electrons[42], and thermally activated higher bands may similarly serve as reservoirs of states that do not contribute to transport[43]. As a result, the charge-carrier density in the conduction band can vary without the sacrifice of overall charge neutrality. In fact, the measured STS (Fig. 2e) shows that $E_F$ does not shift with respect to the edge of the conduction band $CB_1$, i.e., lowest LL, up to 5 T. A variable particle number in the conduction band furthermore agrees with the observed Hall response that does not show the smooth behavior $\rho_{xy} = B/ne$ (which would be expected if particle number were fixed). We find that the behavior of ZrTe$_5$, up to fields of several Tesla is in fact well-described by a model including only the itinerant electrons, but keeping $E_F$ fixed.

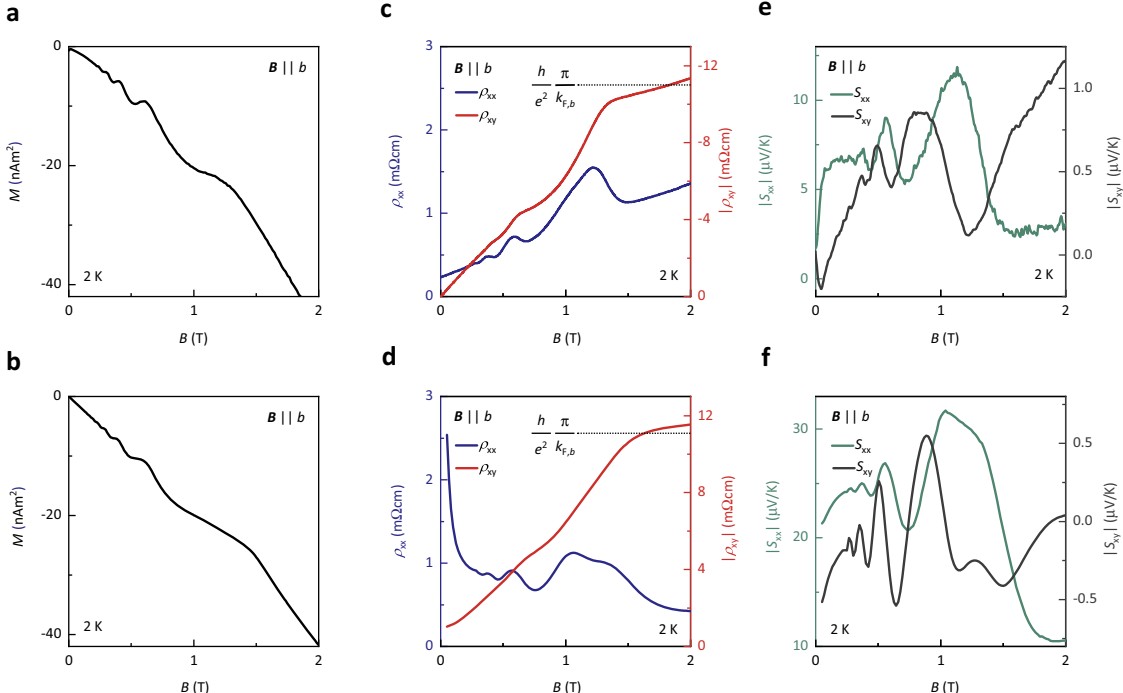

**Fig. 4 Comparison of the experimental and theoretical response of ZrTe$_5$ to magnetic fields $B$. a** Measured and (**b**) calculated magnetization $M$ as a function of $B$ applied along the $b$-axis of Sample G at 2 K. The magnetization plotted in (**b**) is computed numerically from the derivative of the free energy. **c** Measured and (**d**) calculated longitudinal $\rho_{xx}$ (left axis) and Hall resistivity $\rho_{xy}$ (right axis) as a function of $B$ applied along the $b$-axis of Sample A at 2 K. The calculations are done with a quantum lifetime $\tau_Q = 3.064$ ps at a temperature of 2 K. **e** Measured and (**f**), calculated Seebeck coefficient $S_{xx}$ and Nernst coefficient $S_{xy}$ as a function of $B$ applied along the $b$-axis of Sample A at 2 K. The numerical data in (**f**) are calculated for the same parameters as the data shown in (**d**). $\rho_{xx}$ and $\rho_{xy}$ in (**d**) and $S_{xx}$ and $S_{xy}$ in (**f**) are computed from linear response with a fixed chemical potential using the band structure parameters extracted from the experiments.

## Discussion

We now address the expected generality of our results. Our model provides a way to derive and understand the Hall conductivity of a genuine 3D electron systems from the conductance quantum scaled by a characteristic length when only the lowest few Landau bands are occupied. This length scale, however, does not relate to a 2D spatial confinement, but rather to an intrinsic momentum vector of the 3D electron system, which is given by the electronic band structure—a result that is also obtained for a stack of coupled quantum Hall layers[16]. Therefore, we expect quasi-quantized features in the Hall conductivity to be observed in the quantum limit of generic 3D metals and semimetals, regardless of the precise band structure, Fermi level, or purity as long as the particle number of conduction band electrons is not strictly preserved. In fact, our results are consistent with previous reports on ZrTe$_5$ samples that exhibit different $E_F$[44–46]. However, the shape of the quasi-quantized features in $\sigma_{xy}$ will depend on fine details of the 3D Landau-level spectrum. Our findings promise to explain the often puzzling plateaus appearing in Hall measurements in many other low charge-carrier semimetals such as NbP[47] or HgSe[48] and in slightly doped semiconductors such as InAs[42,49] and InSb[50,51], to name a few.

In summary, we have shown that ZrTe$_5$ exhibits quasi-quantized plateaus in the Hall resistivity that scale with $\frac{2\pi}{2k_{F,b}}\frac{h}{e^2}$ in the quantum limit due to an interplay of its Dirac band structure and low charge-carrier density. Our model can be directly applied to Landau quantization of generic 3D band structures, and also to the 3D anomalous Hall and Spin-Hall systems. Our findings establish the Hall effect in ZrTe$_5$ as a truly three-dimensional relative of the quantum Hall effect in 2D systems, and a prime candidate for the observation of relativistic chiral surface states[18,19].

## Methods

**Sample synthesis and preparation**. High-quality single-crystal ZrTe$_5$ samples were synthesized with high-purity elements (99.9999% zirconium and 99.9999% tellurium), the needle-shaped crystals (about $(0.1 \times 0.3 \times 20)$ mm$^3$) were obtained by the tellurium flux method. The lattice parameters of the crystals were structurally confirmed by single-crystal X-ray diffraction. Prior to transport measurements, Pt contacts were sputter deposited on the sample surface to ensure low contact resistance. The contact geometry was defined using Al hard masks. Prior to Pt deposition, the sample surfaces were Argon etched and a 20-nm Ti buffer layer was deposited to ensure good adhesion of the contacts. Deposition was conducted using a BESTEC UHV sputtering system. This procedure allowed us to achieve contact resistance of the order of 1–2 Ohm.

**Sample environment**. The pulsed magnetic field experiments up to 70 T were carried out at the Dresden High Magnetic Field Laboratory (HLD) at HZDR, a member of the European Magnetic Field Laboratory (EMFL). All transport measurements up to ±9 T were performed in a temperature-variable cryostat (PPMS Dynacool, Quantum Design), equipped with a dilution refrigerator insert and a horizontal rotator.

**Electrical and thermoelectric transport measurements**. To avoid contact resistance, only four-terminal measurements were carried out. The longitudinal $\rho_{xx}$ and Hall resistivity $\rho_{xy}$ were measured in a Hall-bar geometry with standard lock-in technique (Zurich instruments MFLI and Stanford Research SR 830), with a frequency selected to ensure a phase shift below 1 degree—typically in the range of $f = 10$–$1000$ Hz across a 100 kΩ shunt resistor. In addition, some samples were measured, using a Keithley Delta-mode resistance measurement setup for comparison. In both measurement modes, the electrical current was always applied along the $a$-axis of the crystal and never exceeded 10 μA in order to avoid self-heating. Thermoelectric measurements were performed using the same electrical contacts as for electrical transport measurements. In order to supply a substantial temperature gradient across the sample despite its high thermal conductivity, the sample was semi-suspended with a heater attached to the free-hanging end. In addition, a set of two Cernox Cx-1030 thermometers were attached to the sample in order to measure the temperature gradient. In order to obtain thermoelectric data within the linear response regime, the applied gradient was kept at <10% of the sample temperature. The thermal voltage was measured using a Keithley 2182 A nanovoltmeter. In order to avoid the influence of parasitic thermal voltages on the cryostat cables, a background measurement without applied power was carried out both for Seebeck and Nernst measurements at all measured fields and temperatures.

**Ultrasound propagation measurements**. Ultrasound measurements in pulsed magnetic fields up to 10 Tesla were performed using a phase-sensitive pulse-echo technique. Two piezoelectric lithium niobate (LiNbO$_3$) resonance transducers were glued to opposite parallel surfaces of the sample to excite and detect acoustic waves. The transducer surfaces were polished using a focused Ion beam in order to ensure that the transducer attachment surfaces were smooth and parallel. The longitudinal and transverse acoustic waves were propagated along the $a$-axis with the transverse polarization vector along the $c$-axis. Relative sound-velocity changes $\Delta v/v$, and sound attenuation $\Delta \alpha$, were measured for field applied along the $b$-axis. The longitudinal and transverse ultrasound propagation were measured at 28 and 313 MHz, respectively.

**Magnetization measurements**. Magnetization measurements were conducted in a standard Quantum Design VSM MPMS equipped with a 7 Tesla superconducting magnet. For measurements, the samples were attached on quartz sample holders and glued using a small amount of GE-varnish. In order to avoid parasitic contributions in magnetic measurements at small fields where the magnetic response of ZrTe$_5$ is small, the background magnetization of the quartz holder together with the adhesive was measured and subtracted from the data.

**In-field single-crystal X-ray diffraction**. In-field single-crystal X-ray diffraction measurements have been performed at the Petra III P21.1 beamline at DESY (Hamburg, Germany). Measurements were performed in a standard cryostat equipped with a 10 Tesla horizontal superconducting magnet. For the measurements, the sample $b$-axis was aligned along the field direction, $c$-axis was in the scattering plane while $a$-axis was vertical. In order to detect new satellite peaks, several reciprocal-space directions were scanned both at 0 and 2 Tesla. Namely, we have performed $k$-scans in the low-Q range $[(0, 0, 0) – (0, 4, 0)]$ and, at high-Q range $[(0, 12, l) – (0, 16, l)$ with $l = 0, 0.5, 0.75, 1$, and $(0, 16, l) – (0, 20, l)$ for $l = 0, 1]$ where the X-ray structure factor is expected to be stronger, For background-free measurements, we have used a CdTe Amptek point detector with a combination of a Ge-gradient Si analyzer and a 101.7 keV incident energy beam.

**Raman spectroscopy**. ZrTe5 flakes were exfoliated by using the scotch tape method and transferred on a SiO$_2$ substrate. The low-temperature magneto-Raman measurements were performed using the attoRaman system and attoLIQUID 2000 cryostat system (Attocube systems AG, Germany) along with WITec 532-nm laser and a WITec spectrometer (UHTS 300 SMFC, WITec, GmbH, Germany) equipped with 1800 g/mm grating and thermo-cooled CCD. The Raman spectra were integrated for 120 s and averaged over five accumulations.

**Scanning tunneling microscopy**. ZrTe$_5$ crystals were cleaved in ultra-high vacuum (base pressure ~$1 \times 10^{-10}$ mbar) at room temperature and immediately inserted into a cryogenic scanning tunneling microscope operated at $T = 1.9$ K. The relatively weak van der Waals bonding between the ZrTe$_5$ layers offers a natural cleaving plane, resulting in a Te-terminated surface. Scanning tunneling microscopy measurements were acquired using electrochemically etched W tips. Spectroscopic data were measured using a lock-in technique, modulating the bias with 1 mV (r.m.s.) at 733 Hz.

**Theory—the Dirac Hamiltonian**. Ab-initio calculations[10] suggest ZrTe$_5$ to be a Dirac semimetal whose low-energy band structure at zero field can be modeled using an anisotropic Dirac Hamiltonian[14]:

$$H(\boldsymbol{k}) = m\tau_3\sigma_0 + \hbar\left(v_a k_a \tau_1\sigma_3 + v_c k_c \tau_2\sigma_0 + v_b k_b \tau_1\sigma_1\right),$$

with $\tau_i$ ($\sigma_i$) denoting Pauli matrices acting on the orbital (spin) degree of freedom and $k_j$ and $v_j$ denote the components of the momentum vector $\boldsymbol{k}$ and Fermi velocity in the $j$th direction, respectively. $m$ accounts for the zero-field gap. The magnetic field $\boldsymbol{B} = rot\boldsymbol{A}$ enters the Hamiltonian through the orbital effect which is implemented via the usual substitution: $\hbar\boldsymbol{k} \rightarrow \hbar\boldsymbol{k} + e\boldsymbol{A}$, and the Zeeman effect is introduced via $H_Z = -\frac{1}{2}g\mu_B\tau_0\boldsymbol{\sigma} \cdot \boldsymbol{B}$. Here, $\boldsymbol{A}$ is the vector potential, $g$ is the Landé g-factor and $\mu_B$ is the Bohr magneton. The spectral function for the corresponding $\nu$-th Landau band at b wave number $k_b$ is:

$$\rho_{\nu,k_b}(\epsilon) = -\frac{1}{\pi}\text{Im}\left(\epsilon - E_\nu(k_b) + i\frac{\hbar}{2\tau_Q}\right)^{-1},$$

with the quantum lifetime $\tau_Q$. For the calculation of the electric- and thermo-electric response functions, we have used the parameters: $m = 10$ meV, $g = 10$, $v_a = 116392 \frac{m}{s}$, $v_b = 15340 \frac{m}{s}$, $v_c = 348875 \frac{m}{s}$, $\mu = 12.745$meV, $\tau_Q = 3.064$ ps.

**Linear response theory—electric and thermoelectric transport**. Electric transport is calculated in linear response via the Kubo formula. The longitudinal and Hall conductivities are obtained from the respective current-current correlation functions. The Green's functions entering these equations contain an imaginary

self-energy to account for the scattering-induced lifetime of quasiparticles. The thermoelectric conductivity tensor $\hat{\epsilon}$ is obtained from the zero-temperature electrical conductivity tensor $\hat{\sigma}$ via integration as

$\hat{\epsilon} = -\frac{1}{|e|T} \int_{-\infty}^{\infty} d\epsilon (\epsilon - \mu) \left(-\frac{\partial n_F(\epsilon - \mu)}{\partial \epsilon}\right) \hat{\sigma}(T = 0, \mu = \epsilon)$. The thermopower tensor $\hat{S}$, which contains the Seebeck and Nernst coefficients, is calculated via $\hat{S} = \hat{\sigma}^{-1} \hat{\epsilon}$.

**Partition function theory—magnetization**. The magnetization $M$ is calculated as the derivative of the free energy $F$ with respect to the magnetic field, $M = -\frac{1}{V} \frac{dF}{dB}$, where $V$ is the volume. The free energy is in turn defined via the canonical partition sum. Here we have neglected Landau-level broadening ($\tau_Q \to \infty$), but have otherwise used the same parameters as described above (see "Theory—the Dirac Hamiltonian").

**Theoretical model for phonon velocity renormalization**. We consider a longitudinal acoustical phonon that propagates along the magnetic field direction (parallel to the b-axis). In the long wavelength limit, the free phonon energy is given by $\hbar\Omega_q = v_s q$, where $v_s$ is the sound velocity and $q$ is the phonon momentum. Electron–phonon interactions induce a self-energy contribution $\Pi^R(q, \omega = \Omega_q)$ to the retarded phonon propagator. Its real part corresponds to the energy correction due to the electron–phonon interactions,

$\mathrm{Re}\Pi^R(q, \omega = \Omega_q) = \hbar\Delta\Omega_q = \Delta v_s q$[52]. Evaluating the self-energy in a random-phase approximation at $T = 0$ and for finite Landau-level broadening $\Gamma > 0$ yields:

$$\frac{\Delta v_s}{v_s} = \lambda B \sum_\nu \int_0^\infty dk_b \rho_{\nu, k_b}(\mu),$$

where $\rho_{\nu, k_b}$ is the spectral function (see section 1.3) of the $\nu$-th Landau band at b wave number $k_b$, and $\lambda = -D_1^2 |e|/(4\pi^2 \rho v_s^2 \hbar) = -1.139 \times 10^{-35} \frac{Jm}{T}$ is a fitting parameter. To account for sample-specific differences in the electronic band structure, we have chosen slightly different parameters in the effective Dirac Hamiltonian. Namely, we have changed the mass parameter to $m = 9.5$ meV, the g-factor to $g = 7$, the Fermi velocity b-component to $v_b = 16486$ m/s and the quantum lifetime to $\tau_Q = 0.658$ps in the calculation of the phonon velocity renormalization. The parameter $\lambda$ is used to approximate the effective electron–electron interaction strength $g_0 = -4\pi^2 \lambda v_s \hbar^2 ((2k_{F,b})^2 + \kappa^2)^2 /|e|$ for electrons at the Fermi level, $k_z = \pm k_{F,b}$, from the very recent work of Qin et al.[37]. Here, $\kappa = \sqrt{|e|^3 B/(4\pi^2 \epsilon \hbar^2 v_b)}$ is the inverse Coulomb screening length with the dielectric constant $\epsilon/\epsilon_0 = 25.3$ evaluated at $B = 1$T.

## Data availability

All data generated or analyzed during this study are available within the paper and its Supplementary Information file. Reasonable requests for further source data should be addressed to the corresponding author.

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

## Acknowledgements

T.M. acknowledges funding by the Deutsche Forschungsgemeinschaft via the Emmy Noether Programme ME4844/1-1, the Collaborative Research Center SFB 1143, project A04, and the Cluster of Excellence on Complexity and Topology in Quantum Matter ct. qmat (EXC 2147). C.F. acknowledges the research grant DFG-RSF (NI616 22/1): contribution of topological states to the thermoelectric properties of Weyl semimetals and SFB 1143. P.M.L., G.G., and Q.L. were supported by the US Department of Energy, Office of Basic Energy Science, Materials Sciences and Engineering Division, under contract DE-SC0012704. J.W. acknowledges support from the DFG through the Würzburg-Dresden Cluster of Excellence on Complexity and Topology in Quantum Matter ct:qmat (EXC 2147, project-id 39085490), the ANR-DFG grant Fermi-NESt, and by Hochfeld-Magnetlabor Dresden (HLD) at HZDR, member of the European Magnetic Field Laboratory (EMFL). J.G. acknowledges support from the European Union's Horizon 2020 research and innovation program under Grant Agreement ID 829044 "SCHINES". We acknowledge DESY (Hamburg, Germany), a member of the Helmholtz Association HGF, for the provision of experimental facilities. Parts of this research were carried out at beamline P21.1 at PETRA III. Y.S. acknowledges funding from the Swedish Research Council (VR) with a Starting Grant (Dnr. 2017-05078) as well as Chalmers Area Of Advance-Materials Science.

## Author contributions

S.G and J.G. conceived the experiment. The single crystals were grown by G.G. The basic structural and transport properties of bulk crystals were measured and studied by P.M.L. and Q.L. S.G., R.W., A.M., and C.F. fabricated the final transport devices. S.G. and R.W. performed the transport experiments. S.G., M.K., D.G., S.Z., and J.W. performed ultrasound propagation measurements. S. D., F. K., and P.S. performed the Scanning Probe Spectroscopy. K. C. and A. S. carried out the Raman experiments. S.S.P.P. supervised the Scanning Probe Spectroscopy and the Raman experiments. S.G. and T.F. performed high field transport measurements. S.G., Y.S., M.Z., and O.I. prepared and executed the in-field X-ray diffraction experiments. S.G., M.B., P.S., and R.K. carried out the magnetization measurements. T.E. and T.M. provided the theoretical model of the three-dimensional quantum Hall effect. S.G., T.M., T.F., and J.G. analyzed the data. All authors contributed to the interpretation of the data and to the writing of the manuscript.

## Funding

## Competing interests

The authors declare no competing interests.
