## [Peer Review File · Nature Communications]

REVIEWER COMMENTS

Reviewer #2 (Remarks to the Author):

The quantum Hall effect (QHE) reported in 3D electronic system, such ZrTe5 has attracted considerable interest, but its mechanism still remains puzzling. In ref. 14, this 3D QHE is attributed to a magnetic-field-driven Fermi surface instability (i.e., the formation of a charge density wave (CDW)) and the transforming the original 3D electron system into a stack of 2D sheets. The current manuscript carried out systematically studies on similar ZrTe5 samples grown by tellurium flux method as those studied in Ref.14, and all of their experiments, including magnetotransport, thermodynamic, magnetization, scanning tunneling spectroscopy, and Raman spectroscopy, in-field single crystal X-ray diffraction as well as ultrasound propagation, did not present any signatures of a Fermi surface instability. Instead, a direct comparison of the experimental data with linear response calculations based on an effective 3D Dirac Hamiltonian suggests that the quasi-quantization of the observed Hall response emerges from the interplay of the intrinsic properties of the ZrTe5 electronic structure and its Dirac-type semi-metallic character. All of the experimental data are very solid and the suggested new model is also interesting. However, I have a few concerns for considerations.

- 1) The band structures of ZrTe5 crystal are very sensitive to the sample quality, this is why there are so many arguments previously on the experimental work in ZrTe5. The authors claim that the samples are very similar to those used in ref. 14 just depending on the R-T curve with peak near 100 K. Did the authors make measurements on ZrTe5 samples with a resistance peak near 145 K (e.g., in ref.13)?
- 2) The authors claim this is due to the interplay of the intrinsic properties of the ZrTe5 electronic structure and its Dirac-type semi-metallic character, why the quantum Hall plateau in quantum limit is not presented in other ZrTe5 crystals? What is the origin?
- 3) In Fig.S10,11,12, the Hall plateau in the quantum limit is not flat, it shows a slop. It looks more like the conventional quantum oscillations. Is the slope change at quantum limit in R_{xy} from the magnetic field-induced change of charge carrier density?
- 4) I notice that, in Fig. S10,11,12,13,15, both the S_{xx} and S_{xy} approach zero at quantum limit. But the theoretical calculation in Fig.S13,14 show deviation from the zero at quantum limit though the shape of the oscillation is very similar. To my knowledge, both S_{xx} and S_{xy} approach zero at quantum limit may indicate much interesting physics. Recently, Zhang et al (PRL 123, 196602 (2019)) has reported the zeroth Landau band of ZrTe5 with R-peak near 145 K, but its thermodynamic behavior is very similar. How do you consider your model with their data? The author should make more comments and explanation on this point because Zhang et al has suggested different model to understand their S_{xx} and S_{xy} .

Reviewer #3 (Remarks to the Author):

The authors made great efforts to address the points raised in my report and added several key

measurements in the new version of the paper, including the longitudinal resistivity in a much lower temperature, Scanning Tunneling Spectroscopy data as well as the Raman scattering data. The quality of the revised version is then greatly improved and now I agree to accept it for publication.

Reviewer #2 (Remarks to the Author):

Dear Referee 2,

thank you for the elaborate and detailed review. We find your appreciation of our work “*All of the experimental data are very solid and the suggested new model is also interesting.*” very gratifying. In the following, we would like to address the points that were raised in the review in the same sequence:

1) *The band structures of ZrTe₅ crystal are very sensitive to the sample quality, this is why there are so many arguments previously on the experimental work in ZrTe₅. The authors claim that the samples are very similar to those used in ref. 14 just depending on the R-T curve with peak near 100 K. Did the authors make measurements on ZrTe₅ samples with a resistance peak near 145 K (e.g., in ref.13)?*

The similarity of our samples to those used in ref. 14 stems not only from the comparison of the R-T curves. To make sure our results would be directly comparable to ref 14 we have made sure to use only ZrTe₅ crystals grown in the same growth setup and conditions as those used in ref.14. The similarity of materials used in both studies is in addition confirmed by comparison of results of the analysis of quantum oscillations and the low field Hall Effect. We find that our samples have Fermi wavevectors and velocities, cyclotron masses, carrier concentrations, mobilities in good agreement to those reported in ref. 14.

In addition to sample quality and specimen-specific variations, we believe that another important factor influencing the experimentally observable physics of ZrTe₅ is the respective sample's Fermi level. This factor is, in our view, not discussed prominently enough, especially because the literature involves both bulk samples and exfoliated devices. Due to the mechanical fragility of ZrTe₅, exfoliation can induce a significant amount of defects in the crystal lattice compared to the pristine bulk samples. Moreover, in very thin flakes (thickness of the order of the Fermi wavelength), quantum confinement effects may become important changing the band structure of ZrTe₅ dramatically compared to bulk samples (Scientific Reports 5,7898 (2015)).

With this prelude on sample variations in mind, let us now answer the referee's thoughtful question. Out of the large number of samples (20+ samples) that we analyzed, none showed a resistance peak 145K. Still, much of the physics we observe in our samples is consistent with data reported for bulk samples with the peak position located at 145K (Guolin Zheng *et al* PRB 96, 121401(R) (2017)). Since our model is as simple as it is generic, there is no reason to doubt that it is appropriate also for ZrTe₅ samples with a larger Fermi surface at low temperatures, such as indicated by the higher temperature of the Lifshitz transition in ZrTe₅. To clarify on that point, we have now added a corresponding sentence into the appropriate position of the manuscript.

“Therefore, we expect quasi-quantized features in the Hall conductivity to be observed in the quantum limit of generic 3D metals and semimetals, regardless of the precise band structure, Fermi level, or purity as long as the particle number of conduction band electrons is not strictly preserved. In fact, our results are consistent with previous reports on ZrTe₅ samples of different Fermi level. (PRB 96, 121401(R) (2017), PRL 123, 196602 (2019), Scientific Reports, 8, 5125 (2018)) However, the shape of the quasi-quantized features in σ_{xy} will depend on fine details of the 3D Landau level spectrum.”

2) The authors claim this is due to the interplay of the intrinsic properties of the ZrTe₅ electronic structure and its Dirac-type semi-metallic character, why the quantum Hall plateau in quantum limit is not presented in other ZrTe₅ crystals? What is the origin?

The referee is entirely correct in pointing out that not all past reports of Hall measurements in ZrTe₅ detail quasi-plateaus. While we are convinced that the observation of quasi-quantized plateaus in the Hall resistivity of Dirac materials should be a rather generic feature, it is by no means fundamentally enforced to appear. There is for example no symmetry enforcing the existence of these plateaus, and they are not topologically protected. We merely point out that they are reproducibly present in a considerable number of samples for a number of compounds, even taken into account the sample-specific variations discussed in point 1.

We propose that the quasi-quantized plateaus appear most likely if the following conditions are fulfilled:

First, the samples have to be very clean in the sense that Landau level broadening does not overshadow the effect. Lattice defects can become particularly problematic in ZrTe₅, because it is an extremely soft material and, hence, stress applied during sample preparation can easily induce defects such as dislocation (see an exemplarily test experiment below). This is especially important during device processing i.e. exfoliation. Moreover, flux-grown samples have proved to tend towards the formation of in-grown twins and tellurium defects.

In response to this comment, we have now exemplarily tested the impact of strain on the transport coefficients of ZrTe₅. To do this, we have selected a nice crystal of ZrTe₅ and measured its Hall and longitudinal resistance at 1.8 K. Subsequently, the sample was taken out of the cryostat, was bent by around 5 degrees and was then bent back. After this procedure, the now potentially “injured” sample was cooled down again and remeasured at 1.8 K. During the whole procedure, the electrical contacts on the sample remained fixed. The results of this test seen in the figure below:

The mechanical bending changes the appearance of the data. At low fields the Hall data overlaps well reflecting that mechanical damage does not change the carrier density, but at high fields it slightly shifts the magnitude of the Hall signal. In addition mechanical damage, significantly increases the longitudinal resistance and slightly suppress the amplitude of quantum oscillations which is in contrast to the Hall resistance much less susceptible to changes in the scattering time.

Our interpretation of this observation is that putting mechanical strain on the sample introduces lattice defects. Such defects act as scattering centers contributing to the Landau level broadening.

Sample A was a rather thin specimen (although still thick enough to be considered as “bulk” – more than 300um) and we believe that the process of mounting it in the setup for thermoelectric measurements added mechanical strain that is now reflected in the rather smeared out appearance of the Hall resistance. We note, however, that the magnitude of the Hall *conductivity* does not change in the quantum limit – in agreement with our theory.

Second, there has to be a mechanism responsible for keeping the chemical potential roughly constant. This could be due to the presence of another band close above the Fermi level (as suggested in arXiv:2011.01952) or presence of defects that soak up carriers in an applied magnetic field (as outlined in ref 42 for the case of InAs).

Third for soaking up enough carriers to fix the Fermi level effectively, the charge carrier density has to be low enough.

Having said this, the plateaus in the Hall effect of ZrTe₅ have been seen at least in 2 other datasets with a Fermi level different than ours (Scientific Reports volume 8, 5125 (2018) and PRB 96, 121401(R) (2017)). In addition, similar effects have been seen in HfTe₅ (Nat. Comm. volume 11, 5926 (2020) and Phys. Rev. B 101, 161201(R)). We consider this as evidence that the effect reported in our manuscript is universal for electronic systems with a low-density.

3) In Fig.S10,11,12, the Hall plateau in the quantum limit is not flat, it shows a slop. It looks more like the conventional quantum oscillations. Is the slope change at quantum limit in Rxy from the magnetic field-induced change of charge carrier density?

Indeed, in contrast to sample B, the plateau of sample A (S10, S11) is not so flat. This qualitative difference is to some extent natural, since – as mentioned in the response to the referee’s previous question– the features appearing in the Hall effect, including the slope of the plateau in the quantum limit, crucially depend on the pinning of the chemical potential and Landau level broadening. In that sense, plateaus in the Hall effect of 3D materials are essentially “conventional quantum oscillations” and the change of slope of Rxy in the quantum limit stems from a field-induced change in the carrier density.

4) I notice that, in Fig. S10,11,12,13,15, both the Sxx and Sxy approach zero at quantum limit. But the theoretical calculation in Fig.S13,14 show deviation from the zero at quantum limit though the shape of the oscillation is very similar. To my knowledge, both Sxx and Sxy approach zero at quantum limit may indicate much interesting physics. Recently, Zhang et al (PRL 123, 196602 (2019)) has reported the zeroth landau band of ZrTe5 with R-peak near 145 K, but its thermodynamic behavior is very similar. How do you consider your model with their data? The author should make more comments and explanation on this point because Zhang et al has suggested different model to understand their Sxx and Sxy.

On a larger scale, the Seebeck voltage in measurements indeed may seem to become zero in the quantum limit (Figures S10 and S12). However, a zoom into those datasets around the quantum limit shown in figures S11 and S13 reveals that while the Seebeck voltage is indeed suppressed, it remains in fact finite and of the same order as the zero-field value. The calculations presented in Fig.4f are in good agreement with the measurement value on that scale.

Similar arguments apply to the work of Zhang et al. Hence, our thermoelectric measurements and calculations seem to be in qualitative agreement with their data al albeit due to the different Fermi

level the appearance of our quantum oscillations seems slightly different due to a weaker effect of spin splitting at low fields.

On the theory side, the model we employ is very similar to the one by Zhang et al. In particular, both models include the same terms (namely, all symmetry-allowed terms). On a finer level, though, there are two differences, namely w.r.t anisotropy and level broadening.

In contrast to the model of Zhang et al., our model uses anisotropic Fermi velocities. Although it might not have been obvious at the time (2019), measurements of quantum oscillations in ZrTe₅ now reveal quite prominently that this compound harbors a highly anisotropic Fermi surface regardless of the Fermi level (for examples see: Min Wu *et al* 2019 *Chinese Phys. Lett.* 36 067201, ref.14 of our manuscript or figure 1 of our manuscript.). We think that it is important to keep track of this anisotropic behavior. Unfortunately, Zhang et al. only discuss measurements of quantum oscillations with one field orientation, which does not allow to recover the anisotropic components of the Fermi velocities (in the case of anisotropic Dirac fermions, one needs to have at hand all 3 components of the Fermi velocity and relevant effective masses). The data presented by Zhang et al thus is insufficient to determine all parameters in our (we believe more realistic) model.

The different treatment of (an-) isotropy in our view constitutes the main difference between our model and the one proposed in the discussed PRL. In their calculation, we believe that Zhang et al. assumed an isotropic Fermi surface and fixed the Fermi velocity such that the last spin split Landau level would cross the chemical potential at 5 Tesla. Using the band structure parameters that Zhang et al reported, we have calculated both S_{xx} and S_{xy} , see figure below:

The results of the calculation seem in good qualitative agreement with experimental data. However there is some discrepancy with the thermoelectric response calculated by Zhang et al. We attribute the difference in calculated response to the way the Landau level broadening is incorporated into the calculations. In our case we consider a fixed Landau level broadening which we use self-consistently first to calculate the chemical potential as a function of field and then to compute the responses themselves. On the other hand, our understanding of the paper of Zhang *et al* is that the chemical potential is first calculated without the inclusion of Landau level broadening and the broadening is only introduced as a field dependent parameter ($\Gamma \propto \sqrt{B}$) in the calculation of the thermoelectric response. Unfortunately, Zhang et al only states that $\Gamma \propto \sqrt{B}$ without citing any absolute prefactor. It is thus difficult to exactly reproduce their calculation.

Again, we thank you for pointing out the possible issues of sample quality and variation of the Fermi level. We hope that our qualitative analysis and further explanations clarified all your concerns and that you find the slightly modified version of the manuscript to be up to the standard that would merit publication in Nature Communications

Sincerely yours,

Authors

Reviewer #3 (Remarks to the Author):

The authors made great efforts to address the points raised in my report and added several key measurements in the new version of the paper, including the longitudinal resistivity in a much lower temperature, Scanning Tunneling Spectroscopy data as well as the Raman scattering data. The quality of the revised version is then greatly improved and now I agree to accept it for publication.

Dear Referee 3,

We would like again to thank you for your involvement in the review process of our manuscript and constructive criticism towards our experimental efforts.

Sincerely yours,

Authors

REVIEWERS' COMMENTS

Reviewer #2 (Remarks to the Author):

The authors answered most of my concerns though some of their answers are not satisfied. I appreciated the authors great effort on the clarification for the nature of the 3D QHE. I have no more new questions for the manuscript, but I suggest the authors must be very caution because the observed nonsaturation behavior of the S_{xx} and S_{xy} in high field is limited by the field range, it will show a big peak in S_{xx} and then saturate in the more higher field (the recent unpublished new data I noticed somewhere). I suggest to accept the manuscript for publication.